# Change in iceberg calving behavior preceded North Sea ice shelf disintegration during the last deglaciation

James D. Kirkham [1,2] ✉, Kelly A. Hogan [1], Robert D. Larter [1],
Neil S. Arnold [2], Ed Self[3], Ken Games[3], Jeremy C. Ely [4], Chris D. Clark[4],
James D. Scourse[5], Calvin Shackleton [6], Jan Erik Arndt [7],
Claus-Dieter Hillenbrand [1], Mads Huuse [8], Margaret A. Stewart[9],
Dag Ottesen[10] & Julian A. Dowdeswell [2]

Understanding how regime shifts in iceberg calving behavior affect ice shelf stability remains a challenge for numerical models. This is an important question as we consider the fate of the ice shelves that currently buttress the Antarctic Ice Sheet and hold back the bulk of its potential upstream sea-level contribution. Using buried landforms, we demonstrate that ice shelves fringed the former British-Irish Ice Sheet (BIIS) and document their disintegration ~18,000 years ago. The ice shelves produced massive (5–10 s km wide, 50–180 m thick) tabular icebergs until widespread ice shelf break-up shifted the calving regime to smaller bergs; a change that coincided with the collapse of marine-based ice across the central North Sea. We propose that the BIIS reached a climatic threshold around 18 ka which caused massive surface melting of its ice shelves, triggering hydrofracturing of crevasses that ultimately led to their disintegration and likely enhanced ice-retreat rates.

Ice shelves fringe the fast-flowing margins of the modern Antarctic Ice Sheet and provide a buttressing backstress that regulates the rate at which upstream grounded ice flows towards the ocean[1–3]. Observations have highlighted a strong association between ice shelf thinning, sometimes leading to their catastrophic disintegration, and the sustained acceleration and retreat of glaciers in Antarctica[3–6]. However, the number of ice shelf disintegration events cataloged within the satellite era remains limited to about 10, e.g., refs. [4,7–13]. Consequently, the manner in which ice shelves and iceberg calving may respond to future climatic warming remains one of the largest sources of uncertainty in projections of future ice sheet evolution and thus sea-level rise[8,14–17].

Ice shelves are vulnerable to changes in atmospheric and ocean conditions[5,12,18,19], losing mass through both melting (thinning) and iceberg calving. Currently, calving accounts for ~45 % of the mass lost from the Antarctic Ice Sheet but with considerable regional variability[20,21]. Observations and numerical models show that the dominant regime of iceberg calving (i.e., the size, shape and number of icebergs calved), and thus the type of iceberg produced, reflects the dynamics of the parent ice mass and the number of pre-existing fractures in a calving margin cf.[22,23] For example, relatively stable ice shelves sporadically produce gigantic tabular icebergs (100 s of meters thick and 10 s to >100 km long) with decades of quiescence between major calving events[24–26]. By contrast, rapidly retreating grounded tidewater glaciers without ice

[1]British Antarctic Survey, High Cross, Cambridge, UK. [2]Scott Polar Research Institute, University of Cambridge, Cambridge, UK. [3]Gardline Limited, Great Yarmouth, UK. [4]School of Geography and Planning, University of Sheffield, Sheffield, UK. [5]Department of Earth and Environmental Sciences, University of Exeter, Penryn, Cornwall, UK. [6]Norwegian Polar Institute, Tromsø, Norway. [7]University of Natural Resources and Life Sciences, Vienna, Department of Ecosystem Management, Climate and Biodiversity, Spatial and Infrastructure Sciences, Institute of Geomatics, Vienna, Austria. [8]Department of Earth and Environmental Sciences, University of Manchester, Manchester, UK. [9]British Geological Survey, The Lyell Centre, Research Avenue South, Edinburgh, UK. [10]Geological Survey of Norway, Torgarden, Trondheim, Norway. ✉e-mail: jamkir56@bas.ac.uk

shelves produce smaller and more frequently calved icebergs that are a fraction of parent ice-mass thickness[27-29]. In the well-documented Larsen-B Ice Shelf disintegration event on the eastern Antarctic Peninsula in 2002, abundant surface melting led to hydrofracturing and rapid fragmentation into large numbers of narrow ice blocks that toppled and collided with one another[30].

Icebergs produce ploughmarks when their keels contact and move through sediment-covered areas of the seafloor. Ploughmarks are commonly observed in multibeam bathymetric or side-scan sonar images of the modern seafloor[31-33], and buried examples have been imaged within Quaternary sediments on formerly glaciated continental shelves, including the North Sea, using 3D seismic reflection methods[34,35]. Morphological analyses of iceberg ploughmarks provide information on iceberg sizes, and therefore calving styles, thus helping to constrain mass-loss mechanisms[36-38]. By far the most abundant ploughmarks are those formed by small, single-keeled icebergs which typically incise narrow V-shaped grooves arranged in curvilinear, chaotic, and criss-crossing patterns as they drift with ocean currents and tides. In contrast, ploughmarks with broad comb-like, parallel morphologies are much rarer and indicate the presence of gigantic tabular icebergs with multiple keels formed by full-thickness calving from floating ice shelves such as those which currently fringe much of the Antarctic Ice Sheet[32,39]. Consequently, the presence and characteristics of iceberg ploughmarks in the geological record can inform about the geometry (presence or absence of ice shelves), mass-loss behavior, and dynamics of former marine-terminating ice sheets, and, critically, how these change with time if ploughmarks are found in geological units with different ages[32].

Recently, Clark et al.[40] presented a new numerical reconstruction of the growth and retreat of the last British-Irish Ice Sheet (BIIS) that used extensive geomorphological observations and chronological data to constrain numerical model simulations of ice sheet flow between 31 and 15 ka. As a result, the BIIS now provides one of the world's best-constrained records of ice sheet deglaciation that can be used to explore how modern ice sheets may decay and, in particular, to aid our understanding of how specific mass-loss processes may evolve in a warming climate. The reconstructions predicted that ice shelves formerly fringed large portions of the BIIS, particularly during the early stages of deglaciation (Fig. 1a). Despite the numerical model used employing relatively simple parameterizations for ocean melt, grounding-line migration and iceberg calving, its predictions are supported by the presence of grounding-zone wedges (GZWs; submarine glacial landforms associated with ice shelf presence), particularly around the BIIS' western margin[41-45]. Some of the largest BIIS ice shelves are predicted by models in the North Sea (Fig. 1a), yet definitive geomorphological evidence for ice shelf presence in the form of GZWs remains scarce in this region, e.g., refs. 46-48, and sedimentary evidence of potential sub-ice shelf facies requires further investigation[47,49]. Furthermore, whilst single-keeled iceberg ploughmarks have been imaged extensively in the North Sea geological record dating back to over 2.5 Ma[34,35], hitherto there has been no empirical evidence for multi-keeled iceberg ploughmarks to confirm the presence of large ice shelves in this region[47].

The extent to which ice shelves modulated the retreat of the former BIIS can provide insight into their importance in determining the long-term response of modern ice sheets to climate change when ice shelves are lost. In this study, we decipher the detailed morphology of iceberg ploughmarks preserved beneath the seafloor of the central North Sea using high-resolution 3D (HR3D) seismic methods to infer the size and shape of icebergs calved from the BIIS during the last glacial period and early deglaciation. We provide direct observational evidence for the presence of tabular icebergs in the North Sea during the last glacial period and trace their disappearance in the geological record, thereby confirming ice shelf occurrence around the decaying BIIS as well as the timing of their disintegration.

## Results

Inspection of HR3D seismic data in seven different survey areas of the central North Sea reveals an abundance of curvilinear grooves with three distinctive morphologies buried 22–55 m beneath the seafloor in modern-day water depths of 115–148 m (Fig. 2). The majority of the curvilinear grooves are unresolvable using the lower-resolution 3D seismic methods previously used to generate the vast majority of Quaternary reconstructions in this region (Fig. 3). The first class of features consists of V-shaped grooves that are 30–80 m wide, ~2–5 m deep, and are flanked along their length by elongate but low-amplitude densely-spaced ridges up to 2.5 m high (Fig. 2a). The grooves extend for 100s–1000 s of meters and terminate abruptly, sometimes leaving circular pit-like depressions in the palaeo-seafloor. The second class of feature comprises broad, relatively flat-bottomed grooves that exceed 4 km in length, are over 200 m wide, up to 6 m deep and are also flanked by elongate ridges up to 1.5 m high (Fig. 2b). The third feature class consists of multiple low-amplitude densely-spaced grooves aligned in parallel to form a comb-like pattern (Fig. 2c). The individual grooves comprising the larger comb-like arrangements are 0.5–1.6 m deep, ~20–80 m wide and are semi-regularly spaced at distances of ~30–65 m. The total width of the comb-like patterns ranges between 300–2350 m, whilst their overall alignment can abruptly change direction by angles of up to 16° over distances of <200 m (Fig. 2c). These subtle features are commonly overprinted by the narrower V-shaped grooves and cannot be resolved using conventional 3D seismic data alone (Fig. 3).

We interpret the three classes of curvilinear grooves as ploughmarks formed by icebergs impacting the seafloor during drift. The parallel ridges flanking the iceberg ploughmarks are berms formed by the redistribution of sediment as the iceberg keels ploughed the seafloor. The ploughmarks are distinct from other previously described glacial lineations in the region (e.g., mega-scale glacial lineations, flutes, and megaflutes) that were molded on top of subglacial tills and ice-overridden glacimarine sediments that are older than 19.5 ka[50-52] (buried ~35 m below seafloor; b.s.f.). These other glacial lineations have been interpreted using 3D seismic surveys and borehole constraints to have been produced by the fast flow of grounded ice during the Late Weichselian[52]. This is because those lineations are straighter, occur over a wider expanse (10 s of kms in width), lack characteristic berms, and do not change direction as abruptly as the iceberg ploughmarks. As many of the iceberg ploughmarks overprint the Late Weichselian glacial lineations[52], the age of these ploughmarks can be constrained to the last deglaciation of the BIIS following the retreat of grounded ice from the central North Sea. Some HR3D seismic datasets also contain examples of similarly ploughed surfaces buried deeper than the Late Weichselian lineations, suggesting that the formative process responsible for the ploughmarks has recurred multiple times during the Quaternary as the North Sea was repeatedly glaciated.

## Discussion

**Multi-keeled iceberg ploughmarks buried beneath the North Sea**
Variations in ploughmark morphology reflect the type of iceberg that formed them. Narrow, single V-shaped ploughmarks are characteristic of scours produced by small, individual icebergs calved from a marine terminating ice margin that is probably grounded, e.g., refs. 53-55. This morphology of iceberg ploughmark is present in all of the HR3D seismic datasets examined. The second class of ploughmark is broader and has a flatter base (>200 m wide) reflecting the imprint of a blocky iceberg with large dimensions or the grounded corner of a tabular berg, e.g., ref. 56. At between 300–2350 m wide, the subtle multi-keeled grooves which comprise the third class of ploughmark are one to two orders of magnitude wider than the other types observed in the HR3D seismic data and are only observed within the deeper waters of the Witch Ground Basin and west of the Norwegian Channel (Fig. 1a). The distinctive parallel groove morphology of these features reflects

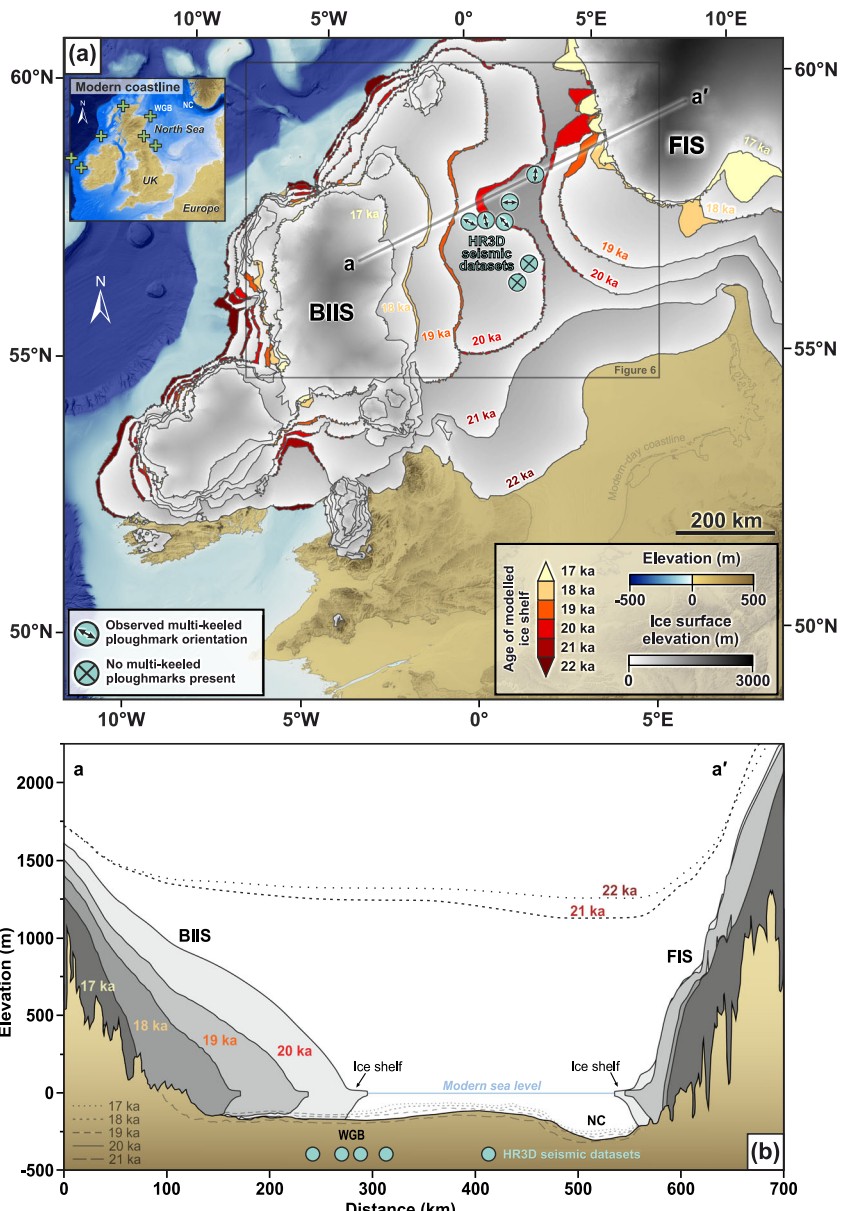

**Fig. 1 | Predicted ice shelves in the North Sea confirmed by the presence of multi-keeled iceberg ploughmarks. a** Model reconstruction of the retreat of the last British-Irish (BIIS) and Fennoscandian (FIS) ice sheets between 22 ka and 17 ka from Clark et al. [40], showing the locations of predicted ice shelves (colored by reconstruction age), and the high-resolution 3D (HR3D) seismic datasets analysed in this study (turquoise circles). The orientations of the multi-keeled iceberg ploughmarks observed within the HR3D seismic datasets (if present) are indicated by arrows. Palaeo topography and water depths at 21 ka are from Bradley et al. [60].

The inset map displays the modern coastline for reader orientation; green "+" symbols mark previously inferred locations of ice shelves surrounding the BIIS from seabed geomorphological evidence[41–44,47,48]. **b** Cross profile of the central North Sea showing the modeled retreat of the BIIS and FIS between 21 ka and 17 ka from Clark et al. [40]. Palaeo topographies (rebounding as the ice retreats) from Bradley et al. [60] are displayed as dashed lines and are in reference to modern sea level. WGB—Witch Ground Basin. NC—Norwegian Channel.

either ploughing by drifting tabular icebergs with massive dimensions (Fig. 4), e.g., refs. 57,58 or incision by drifting sea ice pressure ridges, e.g., ref. 59. These formative mechanisms can be distinguished simply by considering the palaeo water depths at which the ploughmarks were incised, as the keels of sea ice rarely exceed 20 m deep even where pressure ridging has occurred[34,35,59]. Palaeo-sea level reconstructions that account for eustasy and isostasy[60,61] demonstrate that water depths in the region of the central North Sea where the multi-keeled grooves are buried were between ~50–180 m during the growth and retreat of the last BIIS (31–15 ka). This suggests that large icebergs, rather than thinner sea-ice floes, were responsible for the seafloor scouring. Given both the tendency for the ploughmarks to change

direction over short (< 200 m) distances (e.g., Fig. 5a), and the absence of multi-keeled ploughmarks further south in the basin where water depths would have been too shallow to accommodate large icebergs (Fig. 6)[60], we conclude that the grounding of tabular icebergs was the formative mechanism of the multi-keeled grooves observed in the HR3D seismic data (Fig. 5).

As large tabular icebergs require full-thickness calving from ice shelves to form[8], the recorded presence of multi-keeled ploughmarks in the central North Sea provides direct empirical evidence of floating ice shelves in this region during the last glacial period (Fig. 1a). The majority of the multi-keeled ploughmarks are located in the Witch Ground Basin over which the eastern flank of the BIIS retreated as it

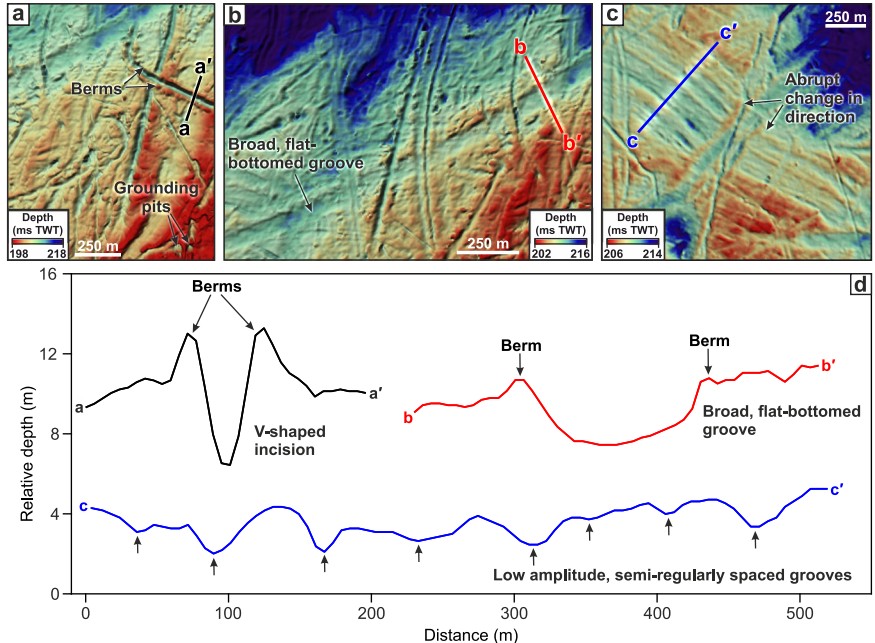

**Fig. 2 | Examples of the three types of buried curvilinear grooves imaged in the HR3D seismic data. a** v-shaped grooves, **b** broad, flat-bottomed grooves, **c** multiple parallel and densely spaced grooves. **d** Representative cross sections of each class of groove.

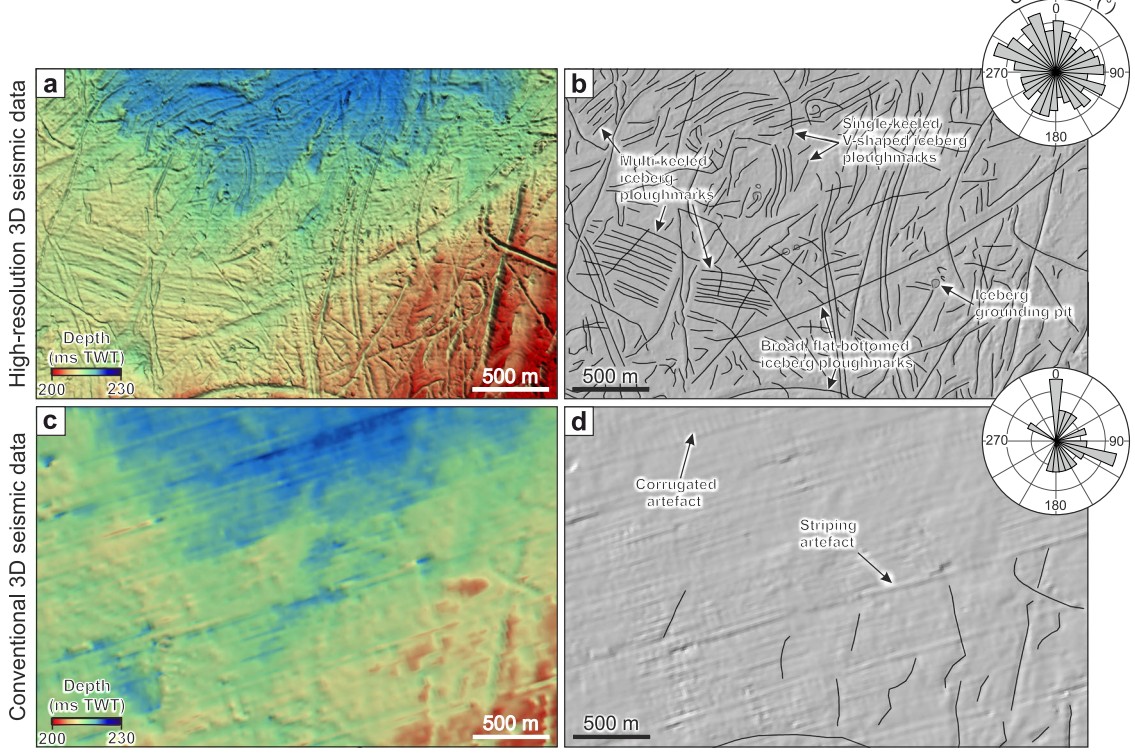

**Fig. 3 | Contrast between buried glacial surfaces mapped using HR3D seismic reflection data compared to conventional 3D seismic reflection data from the same area. a** A glacial surface buried ~30 m beneath the modern-day seafloor mapped using HR3D seismic data and (**b**) geomorphological mapping of iceberg ploughmarks from the surface shown in (**a**), inset with their respective orientations in degrees north. Note the multi-keeled iceberg ploughmarks. **c** The same area and horizon mapped using conventional 3D seismic reflection data. **d** Mapped iceberg ploughmarks visible in (**c**) and their respective orientations. Note the presence of artifact stripes and corrugations in (**c**) which reflect the pattern of the survey lines and aliasing due to gridding at a resolution close to the bin size.

'unzipped' from the Fennoscandian Ice Sheet (FIS) across the central North Sea (Fig. 6)[40,62]. Iceberg ploughmarks observed in the Norwegian Channel off the west coast of Norway are typically oriented north-south[63], implying a northward drift direction towards the shelf break.

This pattern of currents would have likely taken icebergs calved from the BIIS in the Witch Ground Basin eastwards towards the Norwegian Channel before they were transported northwards; this proposed drift direction is supported by an observed shift in ploughmark orientation

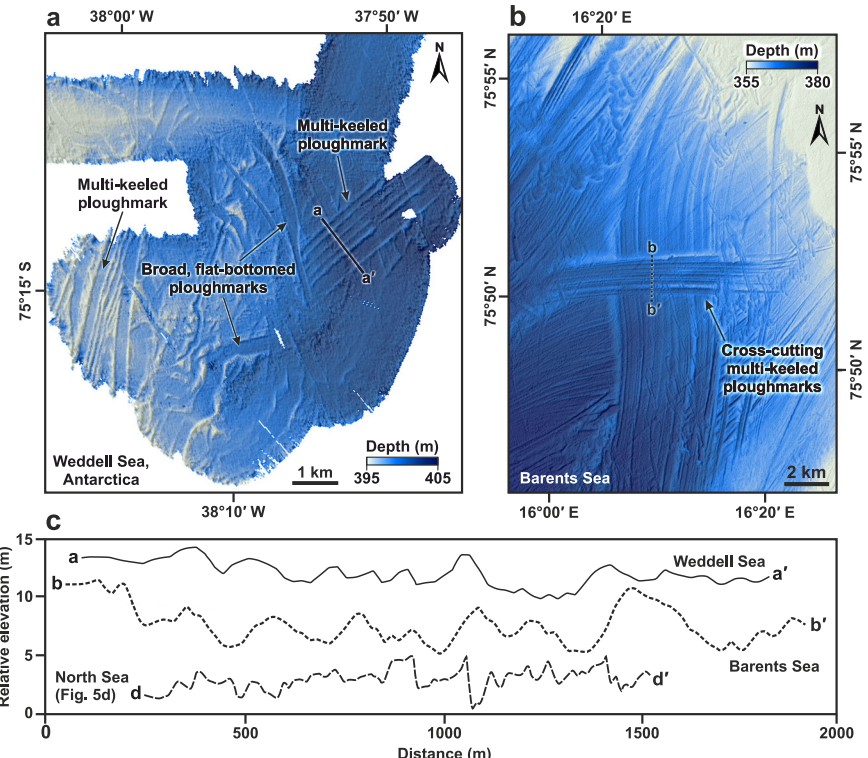

**Fig. 4 | Examples of multi-keeled iceberg ploughmarks in formerly glaciated regions imaged in multibeam bathymetric data. a** Multi-keeled iceberg ploughmarks on the Weddell Sea continental shelf. **b** Cross-cutting multi-keeled iceberg ploughmarks, up to 5 km wide, on the seafloor of the Barents Sea[58]. **c** Cross-sectional profiles of multi-keeled icebergs in the Weddell, Barents and North seas.

from NW-SE in the Witch Ground Basin towards a more NE-SW orientation in the HR3D datasets situated further to the east (Fig. 6). Conversely, the bathymetric sill between the Norwegian Channel and the southern central North Sea would have prevented large icebergs calved from the FIS with drafts exceeding 110 m from drifting towards the Witch Ground Basin[60] (Fig. 6).

At ~50–180 m deep, the inferred keel depths of the tabular icebergs present in the North Sea are analogous in scale to those calved from small fringing ice shelves around modern Antarctica, such as the former Larsen A Ice Shelf, or the extant Brunt Ice Shelf in the NE Weddell Sea Embayment[28]. These ice shelves are typically of the order of ~$10^4$ km² in area and extend ~40–110 km back to the grounding line. The presence of ice shelves in the North Sea had previously only been tentatively suggested from landform mapping[46,47] and numerical model simulations that relied on highly parameterized ocean melt and calving processes, including sub-grid parameterizations of ice shelf movement and grounding-line retreat[40]. These simulations also did not include tidal effects; however, palaeo-tidal simulations demonstrate that this region had the lowest tidal range of all the northwest European ice stream outlets during deglaciation—conditions which would have favored ice shelf formation and stability[64].

### Reconstructed iceberg dimensions

The geomorphological imprints of modern iceberg interactions with the seafloor are challenging to observe in action and, as a result, it is difficult to infer iceberg dimensions from the palaeo-record. One of the most informative examples capable of relating iceberg size to ploughmark dimensions exists on the mid-Norwegian margin where a 2 km wide multi-keeled iceberg ploughmark has been associated with a nearby planed-off debris-flow lobe, suggesting that the iceberg responsible for forming the ploughmarks was at least twice as wide as the seafloor imprint it left behind[32]. In other formerly glaciated regions,

multi-keeled iceberg ploughmarks are commonly 0.3–5 km wide (Fig. 4)[53,57,58,65,66].

Modern tabular icebergs calved from Antarctic ice shelves with thicknesses similar to those inferred to have existed in the North Sea can be larger than 1200 km² in area and more than 50 km wide, such as iceberg A-74 which calved from the Brunt Ice Shelf in 2021 (1270 km²). On the modern seafloor of the Weddell Sea, multi-keeled iceberg ploughmarks up to 1.7 km wide have been observed in contemporary water depths of ~400 m (Fig. 4a)[66]. Tabular icebergs calved in this region between 1984 and 2001 had an average width of 2.4 km, although much larger tabular bergs, such as the 170 × 25 km A-76 (which calved from the Ronne Ice Shelf in 2021), were also produced sporadically[27,67]. Given that present-day water depths on Antarctic continental shelves where multi-keeled iceberg ploughmarks are observed are typically ~300–800 m[68], it is likely that only the largest contemporary icebergs, or potentially even larger icebergs calved from a thicker ice sheet during the last glacial period[69], are responsible for forming the multi-keeled ploughmarks in the Weddell Sea given their sparsity in this region and elsewhere around Antarctica[36]. Furthermore, even the largest multi-keeled ploughmarks observed in Pine Island Trough, West Antarctica, are less than 500 m wide despite the glaciers in this region regularly producing tabular icebergs that are up to 1000 s of km² in planimetric area and 10s–100 km wide[36].

Whilst it is challenging to estimate the original dimensions of icebergs from the seafloor ploughmarks they produce during drift—especially when most contemporary analogs are located in far deeper waters than those present in the North Sea during the last glaciation—a general observation is that multi-keeled ploughmarks tend to be several times smaller than the width of the iceberg which formed them[32,36,53,66]. Using palaeo-water depths[60] to infer the thickness of the North Sea icebergs, assuming hydrostatic equilibrium, e.g., ref. 70, and typical freeboard-width ratios of modern Antarctic tabular icebergs[67],

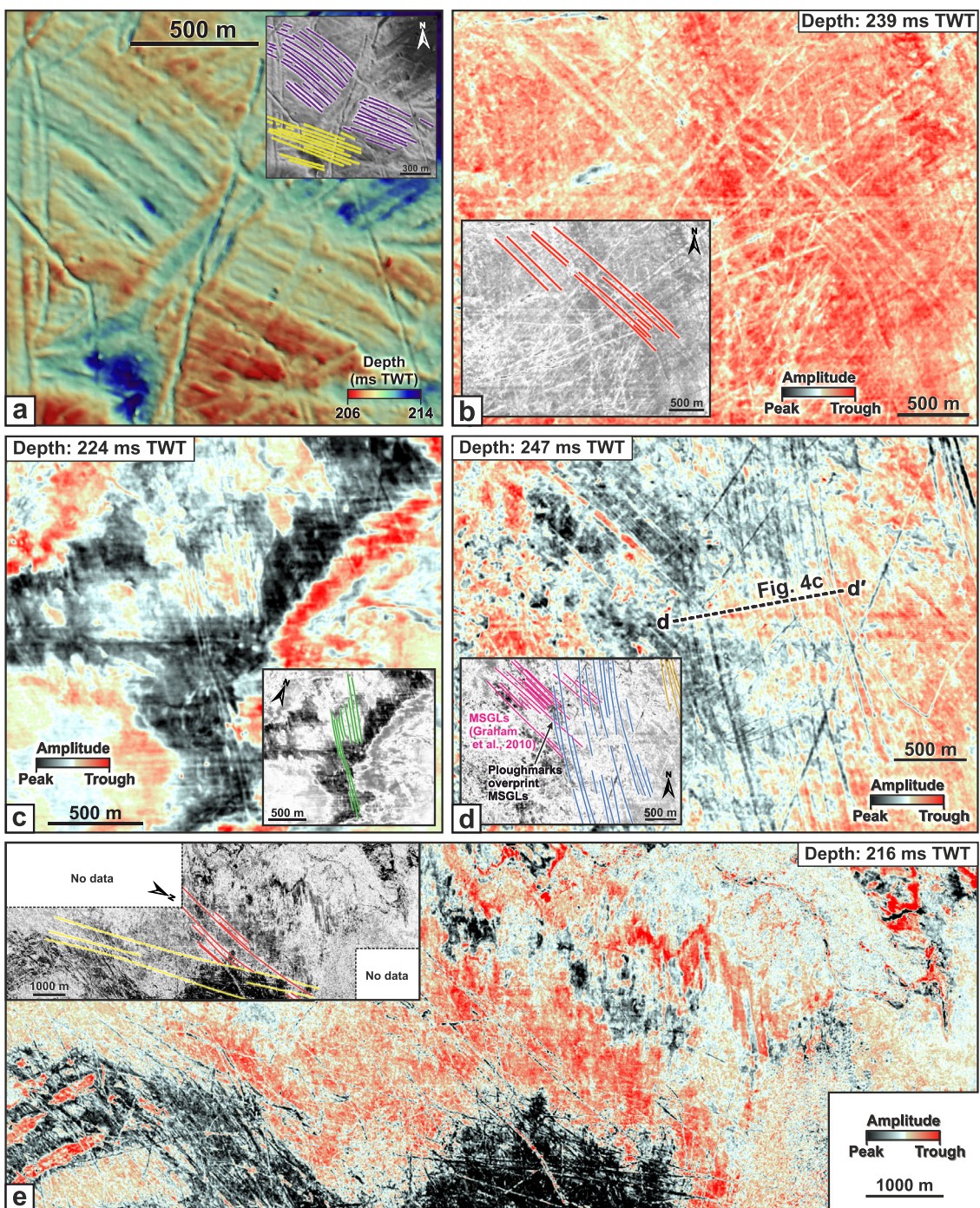

**Fig. 5 | Selected multi-keeled iceberg ploughmarks buried beneath the North Sea, imaged in HR3D seismic data. a** Mapped HR3D seismic surface displaying two flowsets of multi-keeled iceberg ploughmarks cross-cut by smaller single-keeled ploughmarks. **b**–**e** Horizontal timeslices of multi-keeled iceberg ploughmarks buried at different depths in the HR3D seismic data. Note how the multi-keeled iceberg ploughmarks in (**d**) overprint the megascale glacial lineations (MSGLs), interpreted by Graham et al. [52] to be of Late Weichselian (last glacial) age. Ploughmark locations are displayed in Fig. 6.

we estimate that feasible widths of the North Sea tabular icebergs could have ranged from ~500–10,000 m; this range is roughly in agreement with the width of the multi-keeled ploughmarks mapped in the HR3D seismic data. Consequently, given the dimensions of the multi-keeled iceberg ploughmarks observed in the central North Sea and those of their ancient and modern counterparts, the largest tabular icebergs calved from the BIIS during the last glacial period were at least 2350 m wide (the widest observed multi-keeled ploughmark), but their true size was probably significantly larger at 5–10 s km in width.

## Did ice shelf disintegration trigger the retreat of the BIIS from the North Sea?

Modern rates of iceberg calving are primarily set by structural and internal stresses within an ice shelf created as ice flows towards the ice margin[8,12,23]. Increased fracturing or thinning due to substantial changes in ice flow or ocean-induced basal melting can diminish the structural integrity of an ice shelf, potentially resulting in a calving regime transition that may drive rapid changes to terminus position, or make an ice shelf more susceptible to external environmental drivers

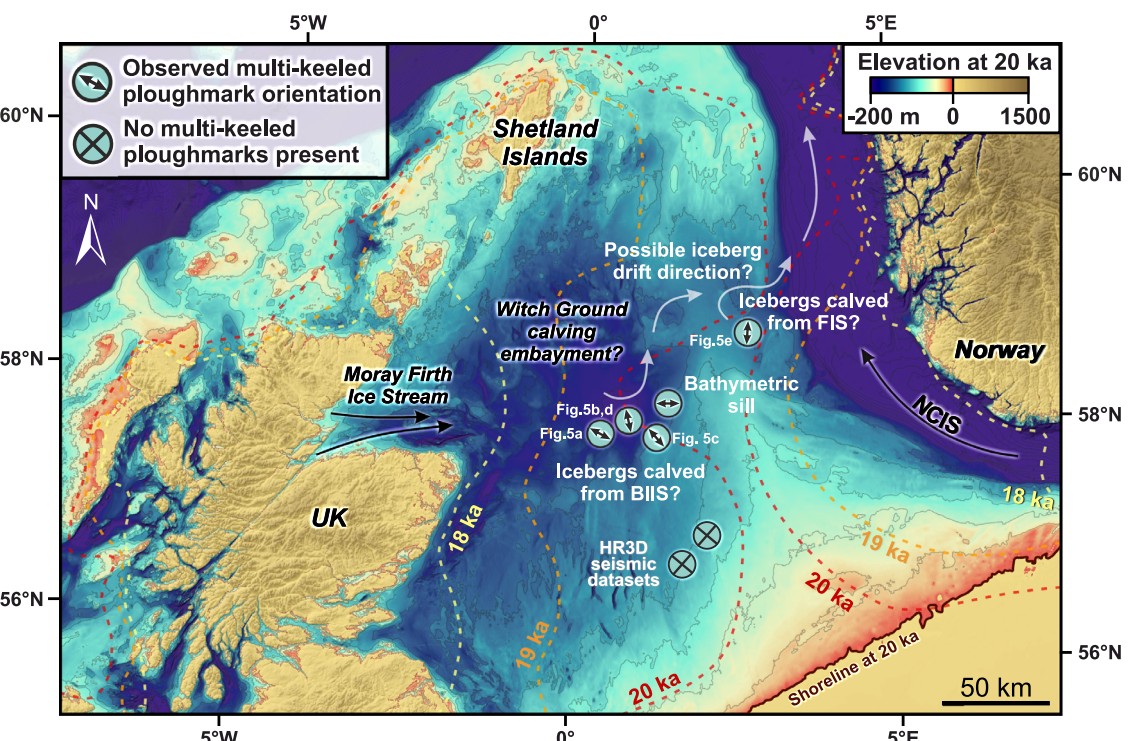

**Fig. 6 | Proposed drift direction of icebergs calved from the BIIS during the last glaciation based on the orientation of multi-keeled iceberg ploughmarks.** Icebergs calved in the Witch Ground Basin would drift eastwards towards the Norwegian Channel before being routed northwards towards the continental shelf edge. The isostatically-corrected bathymetry of the central North Sea at 20 ka is from Bradley et al. [60], with bathymetric depth contours displayed every 25 m. Colored stippled lines represent the modeled ice margins of the last British-Irish (BIIS) and Fennoscandian (FIS) ice sheets at timeslices between 20 ka and 18 ka from Clark et al. [40]. NCIS−Norwegian Channel Ice Stream.

that could precondition or trigger rapid collapse cf. [8,12]. Most rapid disintegration-style events observed to date in the observational era were ultimately triggered by enhanced surface and/or basal melting[4,7,9,10,13,71].

The multi-keeled iceberg ploughmarks observed in the Witch Ground Basin are situated directly above glacial lineations (~35 m b.s.f.) formed by the flow of grounded ice just prior to the retreat of the last BIIS from this part of the central North Sea, an event that has been dated to ~20 ka[51,52]. At shallower burial depths within our data at this location (~12 m b.s.f.), evidence of multi-keeled iceberg ploughmarks disappears and stratigraphic horizons dated largely to 17–14 ka are instead ploughed by smaller single V-shaped grooves; these in turn are buried by postglacial Holocene muds and silts[52,72]. This switch in the type of iceberg ploughmark demonstrates that a calving regime shift from broad parallel comb-like to single iceberg ploughmarks occurred sometime between ~19–17 ka, when the sporadic calving of large tabular icebergs ceased and instead switched to the production of many smaller single-keeled icebergs.

In contemporary settings, similar transitions in iceberg production have been associated with ice shelf collapse through the process of repeated iceberg fracture, detachment, and a corresponding acceleration in mass loss for the grounded portions of marine-based ice sheets[22]. For example, the Larsen B Ice Shelf collapsed at a rate of ~10 km week⁻¹ once the calving of large tabular icebergs ceased and mass was instead removed via the loss of numerous, yet relatively small (<250 m wide), pieces of the ice shelf[8,71]. Once lost, the reduction in buttressing from ice shelves has an almost instantaneous effect on ice flow[3,73], resulting in rapid increases in ice flux over the grounding line that can propagate far inland with significant implications for ice sheet stability[8]. We therefore interpret the regime shift in iceberg production to represent a significant event in the mass loss from the BIIS: the widespread disintegration of its ice shelves in the North Sea between

~19–17 ka. In this broader context, it is notable that there is a coincidence between the timing of disintegration of the BIIS ice shelves and the saddle collapse of the grounded portions of the BIIS and FIS across the North Sea[40](Fig. 1a).

An important question, therefore, is whether ice shelf breakup was the main driver of accelerated grounded ice retreat across the North Sea, or was disintegration merely a symptom of enhanced ice sheet mass loss that was already underway when the ice shelves collapsed? Almost all modern ice shelves serve to buttress inland ice[8], providing a 'safety band' which slows the flow of grounded ice towards the ocean[1,2]. The efficacy of buttressing is significantly reduced, however, when ice shelves are laterally unconfined by the surrounding ice or bedrock[14], as is the case for several East Antarctic ice shelves[2]. The BIIS's ice shelves are modeled to be largest in embayments where the margins of the ice shelves are laterally confined, particularly at 20 ka and 19 ka where grounded ice is pinned around the rim of the Witch Ground Basin (Fig. 6). Accordingly, the former ice shelves fringing the BIIS likely restrained the flow of upstream grounded ice towards several of the BIIS's marine terminating margins. It then follows that any reduction in ice shelf area would reduce buttressing and result in accelerated grounded ice retreat across the relatively smooth low-gradient bed of the central North Sea.

Ice shelves are sensitive to environmental changes in both the atmosphere and the ocean[10,74,75]. Their destabilization can arise via surface thinning through melting, hydrofracture, lake ponding and drainage[4,19], or through weakening from beneath via basal melting, undercutting, grounding line retreat, or in response to increased ocean swell[18,20,30,76]. The thermal limit of extant Antarctic ice shelves has been observed to follow the migration of the −9 °C mean annual air temperature isotherm[11] because above this temperature extensive melt ponds form in summer which lead to large scale ice shelf fragmentation and collapse driven by

hydrofracturing[7]. Recently, however, it has been argued that the melt-over-accumulation ratio plays a more important role than annual mean temperature for setting the threshold of Antarctic ice shelf stability[77]. Although reconstructions of the demise of the BIIS show that retreat initiated at around 22 ka−thousands of years prior to deglacial warming and $CO_2$ rise−changes in ice sheet volume appear to mirror variability in June insolation[40]. This suggests that the earlier start of deglaciation may have been triggered by orbitally induced increasing levels of summer solar radiation that produced greater surface melting[40,78]. Accordingly, it is feasible that the magnitude of surface melting reached a threshold between ~19–17 ka, causing the ice shelves fringing the BIIS to undergo widespread hydrofracture and collapse, accelerating ice losses around its marine margins. A shift in surface meltwater presence has also been linked to the rapid deposition of thick grounding-line proximal glacimarine muds that were deposited in the Witch Ground Basin at around this time[79]. Alternatively, changes to internal ice sheet dynamics, potentially driven by the effects of sea-level rise and/or atmospheric or ocean warming, could also have altered the calving regime through facilitating different rifting patterns and calving frequencies.

Regime transitions in iceberg calving from modern Antarctic ice shelves currently represent some of the largest uncertainties in sea-level rise projections, yet very few examples of complete ice shelf loss have been observed, and humanity is yet to witness the longer-timescale equilibration of an ice sheet to the loss of its ice shelves[8]. Our observations document that such a regime transition occurred previously in the North Sea, with the style of ice shelf calving shifting from the full thickness calving of tabular icebergs to the rapid calving of numerous smaller ice blocks as the BIIS's ice shelves disintegrated −an event that coincided with the collapse of the BIIS's marine-based sectors. The scarce number of ice shelf collapse events that have been documented in the satellite era suggests that similar transitions in iceberg calving regime are often associated with rapid ice shelf loss[4,8].

Such events represent a mere blink of an eye in the long-term lifecycle of an ice sheet and its corresponding sedimentary legacy. Consequently, advances in the resolution and precision of chronological constraints in the North Sea are presently required to decipher whether ice shelf loss triggered the collapse of the marine sectors of the BIIS, or whether disintegration was merely a symptom of wider ice sheet losses that were already underway when its ice shelves collapsed. However, the impacts of such a shift were likely far longer-lasting and more substantial than the duration of the event itself, especially in the case of the relatively flat-bedded areas of the former BIIS. This is because recent work has highlighted how ice shelf thinning, iceberg calving and/or sea-level rise can trigger pulses of very rapid grounding line retreat at rates of at least 2 km yr$^{-1}$ [80,81], and potentially up to 600 m day$^{-1}$ [82], across relatively shallow-gradient (< 0.1°) beds after ice contact with stabilizing topographic high points is lost. As much of the regional gradient of the isostatically rebounded central North Sea also falls below this threshold[60], it is possible that similarly high rates of ice retreat were promoted across the central North Sea as the grounding line unpinned from the rim of the Witch Ground Basin in response to ice shelf disintegration. The North Sea ice shelves documented here may therefore have played a significant role in promoting the stability of the BIIS in response to early deglacial warming. Given that extensive areas of the contemporary Antarctic Ice Sheet are grounded on similarly low or retrograde gradient beds[83], further assessment of how the transition in calving regime and subsequent loss of ice shelves documented by our data preconditioned, or even triggered, the demise of the BIIS may provide crucial insights on future sea-level rise trajectories from the Earth's remaining ice sheets.

## Methods

### High-resolution 3D seismic data

We examined the morphology of the former North Sea seafloor using high-resolution 3D (HR3D) seismic data, which reveal intricate morphological structures of glacial origin that cannot be resolved using conventional 3D seismic methods[84–86]. Seven HR3D seismic datasets were examined in the central North Sea (Fig. 1a) covering a combined area of ~67 km². The data were acquired using two 1200 m-long streamers towed 3 m beneath the sea surface with 96 hydrophone groups at 12.5 m spacing with a response that varied by <±0.5 dB over frequencies between 2–500 Hz, a 6.25 m shot interval and a 1-ms sample rate[87]. The seismic source was two 160-inch[3] sleeve airgun clusters with a 20–250 Hz effective frequency band. Data processing included swell noise attenuation, tide correction, multiple suppression, two passes of velocity analysis run at 250 × 250 m intervals, normal moveout correction and bandpass filtering. The final processed datasets consist of time migrated 3D stacks with a 1 ms sample rate, a 6.25 × 6.25 m bin size, a vertical resolution of ~4 m, and a detection limit along individual reflectors of ~0.5 m[84,88]. Palaeo-seafloor morphology was examined using sequential horizontal timeslices from the HR3D seismic volumes in S&P Global Kingdom software. Each former seafloor was digitized in 3D as an individual seismic horizon. The mapped seismic horizons were converted from two-way travel time (TWT) to depth using a sound velocity of 1850 m s$^{-1}$ [89].

### Mapping of multi-keeled iceberg ploughmarks in the Weddell Sea

Multibeam bathymetric data from the Weddell Sea were acquired from RV *Polarstern* with a 15.5 kHz ATLAS Hydrosweep DS3 system during cruise PS96 in 2015–2016[66,90]. Data were gridded at a 25 m resolution, and iceberg ploughmarks on the seafloor were mapped in ArcGIS software.

## Data availability

The confidential industry datasets analysed in this study are owned by BP, Harbour Energy, Equinor Energy AS, Petoro AS, Aker BP ASA, TotalEnergies EP Norge AS, and TGS. Interpretations made from these data are available from the corresponding author on reasonable request.

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

## Acknowledgements

We thank BP, Harbour Energy, CNOOC, Equinor Energy AS, Lundin Energy Norway AS, Petoro AS, Aker BP ASA, TotalEnergies EP Norge AS and TGS for access and permission to publish images extracted from the HR3D seismic data and the Central North Sea MegaSurveyPlus. S&P Global and Schlumberger are thanked for providing academic seismic interpretation software licenses. J.D.K. was supported by the Natural Environment Research Council (grant NE/L002507/1). K.A.H., R.D.L. and C.-D.H. were supported by the Natural Environment Research Council – British Antarctic Survey Polar Science for a Sustainable Planet programme. J.C.E. was supported by a NERC independent research fellowship (NE/R014574/1). C.D.C. was supported by the European Research Council (PalGlac; Grant Agreement No. 787263). The interpretations made in this paper are the views of the authors and not necessarily those of the license owners.

## Author contributions

J.D.K., K.A.H., R.D.L., J.A.D., and N.S.A. conceived the study. E.S., K.G. and M.H. worked with data owners to gain permission for the use of the 3D seismic data in this project, and C.-D.H., J.E.A., and C.S. provided and

contributed to the interpretation of the multibeam bathymetric data. J.D.K. analysed the 3D seismic data with contributions from K.A.H., R.D.L., J.A.D., E.S., M.H., M.A.S., and D.O. The BRITICE-CHRONO model data were provided by J.C.E., C.D.C., and J.D.S. J.D.K. wrote the initial draft of the manuscript and produced the figures.

## Competing interests

The authors declare no competing interests.
