## [Peer Review file · Nature Communications]

Change in iceberg calving behaviour preceded North Sea ice shelf disintegration during the last deglaciation

Corresponding Author: Dr James Kirkham

Version 0:

Reviewer comments:

Reviewer #1

(Remarks to the Author)

This study shows the first identification of buried Late-Glacial iceberg plough marks from multi-keeled icebergs (i.e. tabular icebergs from ice shelves) in the North Sea. The study first measures the characteristics and dimensions of the plough marks, thereafter interpret their presence in relation to different types of iceberg calving from the ice shelves of the BIIS, and distinguishes that these icebergs are from the BIIS and not from the FIS. This is a very important contribution for our understanding of the glacial dynamics during the deglaciation of the North Sea, in the confluence area between the BIIS and FIS. The data set is impressive, and the interpretation is thorough and well aware of artefacts and alternative causes for the observations.

The results of this novel study will be of high relevance to the large communities working on reconstructions of the last Eurasian ice sheet (based on measured data as well as modeling), and Quaternary seismic stratigraphy in the North Sea region, as well as for the studies of present ice shelves in Greenland and Antarctica. The study is well in line with the established literature and existing records of the North Sea Late Glacial environment (i.e. not controversial), but provides new data with high resolution from areas/depths hard to reach. The HR3D-technique is still new, and this paper will, with its clear and useful results, likely inspire more, similar, studies in the near future.

While the presence of ice shelves in the North Sea had previously been suggested from landform mapping and numerical model simulations, the HR3D measurements done in this study provide more robust and direct evidence which clearly merits publication in Nature Geoscience. Ice-shelf disintegration is a very short time-scale process compared to the lifecycle of an ice sheet. While they have far-reaching consequences of rapid ice loss from the grounded ice, they leave limited evidence of their occurrences. It is easier to document them in the present day, but highly challenging to constrain in paleo timescale. In that regard, the paper provides evidence of the presence of ice shelves and subsequent calving of small ice bergs coincident with the deglaciation of BIIS.

The claims are well supported by the data, by references to relevant literature and by logical reasoning. The methodology, data analysis, interpretation and conclusions are scientifically sound. There is enough detail provided in the methods for the work to be reproduced (apart from a minor detail specified below). The figures are clear, of very high quality, relevant and adequate, with clear labeling (with one exception, see below Fig 1b) and legends.

We strongly recommend this manuscript, with minor revisions, to be published in Nature Communications.

Detailed comments:

L 5: Please name a few of those specific processes that are difficult to capture in the satellite era (examples).

L 87-88: Please add a short statement about what the overlying 35 m of sediments in the investigated area consists of (glacial marine, and Holocene silt/clay, till?), with reference to Graham et al. (2007) and Graham et al. (2010).

L 11: Is there any proxy data or evidence from atmospheric/ climate side that can complement the massive surface melting?

L 19: Comment to the authors /editor about style – using superscript index numbers for references, it looks a bit weird when references are given as examples, “Limited to about 10 e.g., 4,7-11 “. Why not but the “e.g.” in superscript too?

L 24-25: Can you please write a sentence how number of pre-existing fractures in calving margin is related to type of ice produced?

L 65: How do you trace their disappearance? I would like to see this later.

L 93-95: Is this from some literature or found in the present HR3D survey? In case of former, please provide the reference. In case of latter, add some detail on it.

L 108: The comment about palaeo water depths at 20-18 ka is the first mention of the age (apart from the Abstract, but that

should be a standalone text). This is a bit confusing as the discussion about the likely age of the iceberg scours comes later. This statement needs to be preceded by some motivation for the choice of age or be replaced by a more general statement about the likely water depth (e.g. a guess is that the water was deeper than ~20 m (i.e. deeper than sea ice reach) all the time since 30 ka at the locations in question). We also suggest you add a reference to Clark et al., 2022b (<https://doi.pangaea.de/10.1594/PANGAEA.945729>), where the palaeo-bathymetry map datasets are published. It is not clear where you got the numbers ~65-180 m palaeo water depth from. Two references are given (Bradley et al., 2023; Clark et al., 2022) which are correct and relevant in that they describe how the palaeo-bathymetry for the whole BRITICE region was calculated, but the exact information (the palaeo-waterdepth at your location) is not clearly given in any of these papers. L 121-123: "Conversely, the presence of a bathymetric sill [...] would have prevented large icebergs calved from the FIS [...] from drifting towards the Witch Ground Basin". Delete "presence of a", as this gives the impression that "if a sill existed it would have prevented...". The sill is there, evident in the bathymetry. "Conversely, the bathymetric sill [...] would have..." is clearer.

L 166-172: The authors show three different types of ploughmarks in Fig. 2 and associate them with different dimensions of icebergs. While an analogy is provided that big tabular iceberg formation was followed by smaller icebergs, this can be further complemented with the location of the different ploughmarks in the direction of ice sheet retreat. Can you correlate the position of ploughmarks and dimension of iceberg with the retreat of the BIIS? We feel this has been attempted in the first paragraph of the third discussion point, but the evidence for this development is not clear. Please clarify this in the text and by adding appropriate notations/labels to Fig. 6 to show where smaller iceberg scours were identified.

Fig. 1b: The blue colour of the North Sea under the white ice gives the impression that ice was not grounded there at 22-21 ka (which it clearly was). This might be confusing to some readers. I suggest you indicate the present sea level with a thin line across the section and make the North Sea be white (as the ice) with blue hatching (////) below sea level. Also, indicate that the y-axis scale is based on present-day topography.

L173-183: Iceshelf calving can happen irrespective of warming due to changes in stress regime, while warmer climate increase the frequency. For example, due to topographic undulation or some other reason, there can be rift or fracture formation that may lead to calving. This is in particular true for Antarctica and Greenland, but may not be true for BIIS and FIS confluence zone. Basically, ice shelves can break during growing, retreat and intermittent phases. A question in that regard: can you discern that those ploughmarks were due to the icebergs that calved due to warming but not due to "natural" calving or during the growth of ice sheet? Such a clarification would strengthen the text.

L 388, 391, 526, 533, 537: doi:doi: Remove "doi:".

Fig. 5. Please indicate North (with an arrow or in the caption) in the HR3D images. Please add the locations of these in Fig. 6. Why not show the fifth example?

Fig. 6. Please indicate where the HR3D-slices from Fig. 4 and 5 are located in the map (maybe by a small index number next to the arrow-circles). It would be especially interesting to see which ploughmark set that is the easternmost indication in Fig. 6 (possibly calved from the FIS).

Methods: Please include information about what frequency span the hydrophones were sensitive to.

Richard Gyllencreutz and Ankit Pramanik

Reviewer #2

(Remarks to the Author)

Reviewer #3

(Remarks to the Author)

What are the noteworthy results?

The noteworthy results of this study are applying High-resolution 3D (HR3D) seismic data interpretations to the North sea to document and interpret curvilinear grooves with three distinct morphologies to better understand the evolution of the British-Irish Ice Sheet. This new approach using HR3D seismic data allowed the research team to identify three distinct features that are compared to the Barents, and Weddell Sea multi-keeled ploughmarks. Additionally, this data is compared to previous studies in the region to develop a timeline of events, and to better assess the role of ice shelves in promoting stability in the British-Irish Ice sheet during early deglacial warming. Furthermore, these observations have the potential to better assess polar response to warming, particularly in or around Antarctica as there are many similarities between the two environments described by the team.

Will the work be of significance to the field and related fields? How does it compare to the established literature? If the work is not original, please provide relevant references.

This work is significant in that it provides an opportunity to study the transition in calving regime of Northern Hemisphere ice sheets and the loss of ice shelves documented in this study can precondition or even start the demise of an ice sheet, which is critical for understanding how the Antarctic Ice Sheet will respond to warming temperatures. These observations are some of the missing critical pieces needed to build confidence in our projections of future sea level rise from the loss of ice sheets.

Does the work support the conclusions and claims, or is additional evidence needed?

The authors do a good job of supporting their conclusions and claims. Providing evidence of similar features and a comprehensive description of the prior data in this region is organized in a way that steps through the new technique, shows

how it improves on previous seismic observations, and the applies the new data from HR3D seismic to the environment to get a sense of the ice sheet/shelf history in the North sea and compares to other observations in the Barents and Weddell sea.

Are there any flaws in the data analysis, interpretation and conclusions? Do these prohibit publication or require revision?

The authors do an excellent job of presenting their data and interpreting it based on other observations in the North Sea and other environments past and present that have similar features, trying to better understand this complex system.

Is the methodology sound? Does the work meet the expected standards in your field?

The methodology and conclusions seem to build off of the literature and provide evidence for how this new technique is important for a more detailed look at these important settings. My field is not in seismic interpretation, so it is hard for me to fully say, but I do work in the subglacial environment, and I believe their conclusions fit the data and interpretations they presented.

Is there enough detail provided in the methods for the work to be reproduced?

My background is not in seismic interpretations, but it seems that they provide robust references and descriptions for how they collected, processed, and interpreted the data.

Other minor comments highlighted in the text:

Line 18: ice-shelf from my understanding should not be hyphenated. There are times in the text where it is and so I would say for consistency just make sure you are either hyphenating all the ice shelf/shelves or removing it hyphen.

Line 21: ice sheet (similar to comment for line 18)

Line 22: In your call out to ice shelves being vulnerable to changes in atmospheric and ocean conditions you cite a paper on annual mass budget, which uses this statement as well. Might be better here to cite papers documenting atmospheric and ocean conditions leading to ice shelf vulnerability.

Line 28: Seems like there should be some citations here of sporadic tabular icebergs followed by periods of quiescence between calving.

Line 51-52: This sentence seems a bit awkward, so it might be good to rewrite/restructure it.

Line 61-64: The sentence seems a bit awkward. It might be better to say, ".. North Sea using newly captured high-resolution three-dimensional (HR3D) seismic data to infer.."

Line 73: This might be too picky (and you might not have the words/characters) but it would be good to introduce these features. "We identify three classes of features associated with curvilinear grooves from HR3D seismic in the North Sea: the first feature...."

Line 97-99: It would be good to cite some of the work that supports the individual iceberg for these characteristic scours. If they are distinguishing features and it has been observed in other regions/associated with individual icebergs then you should cite those studies like the other classes have specific citations.

Line 135: Add hyphen: palaeo-record

Line 157: Add hyphen: palaeo-water

Line 208: should there be a hyphen?

Line 218: should there be a hyphen?

Version 1:

Reviewer comments:

Reviewer #1

(Remarks to the Author)

2nd Review

The authors have addressed all our comments and questions in the manuscript (both text and figures) in adequate ways, and given very well written motivations for their choices.

Detailed comments:

L. 65 Eustacy -> Eustasy (from greek stasis = standing).

We have no further comments. Well done, interesting work!

Best wishes

Richard and Ankit

Reviewer #2

(Remarks to the Author)

Reviewer #3

(Remarks to the Author)

This revised manuscript enhances a novel and well-crafted study with significant implications for reconstructing the last Eurasian Ice Sheet, past iceberg calving events, and the Greenland and Antarctic research communities. The authors effectively addressed all suggestions from the first round of reviews. I appreciate and recognize the effort the authors put into incorporating the feedback. Each suggestion was thoughtfully considered, and I see no issues with advancing this toward publication. Therefore, I strongly encourage the publication of this manuscript in Nature Communications.

Reviewers 1 & 2:

This study shows the first identification of buried Late-Glacial iceberg plough marks from multi-keeled icebergs (i.e. tabular icebergs from ice shelves) in the North Sea. The study first measures the characteristics and dimensions of the plough marks, thereafter interpret their presence in relation to different types of iceberg calving from the ice shelves of the BIIS, and distinguishes that these icebergs are from the BIIS and not from the FIS. This is a very important contribution for our understanding of the glacial dynamics during the deglaciation of the North Sea, in the confluence area between the BIIS and FIS. The data set is impressive, and the interpretation is thorough and well aware of artefacts and alternative causes for the observations.

The results of this novel study will be of high relevance to the large communities working on reconstructions of the last Eurasian ice sheet (based on measured data as well as modeling), and Quaternary seismic stratigraphy in the North Sea region, as well as for the studies of present ice shelves in Greenland and Antarctica. The study is well in line with the established literature and existing records of the North Sea Late Glacial environment (i.e. not controversial), but provides new data with high resolution from areas/depths hard to reach. The HR3D-technique is still new, and this paper will, with its clear and useful results, likely inspire more, similar, studies in the near future.

While the presence of ice shelves in the North Sea had previously been suggested from landform mapping and numerical model simulations, the HR3D measurements done in this study provide more robust and direct evidence which clearly merits publication in Nature Geoscience. Ice-shelf disintegration is a very short time-scale process compared to the lifecycle of an ice sheet. While they have far-reaching consequences of rapid ice loss from the grounded ice, they leave limited evidence of their occurrences. It is easier to document them in the present day, but highly challenging to constrain in paleo timescale. In that regard, the paper provides evidence of the presence of ice shelves and subsequent calving of small ice bergs coincident with the deglaciation of BIIS.

The claims are well supported by the data, by references to relevant literature and by logical reasoning. The methodology, data analysis, interpretation and conclusions are scientifically sound. There is enough detail provided in the methods for the work to be reproduced (apart from a minor detail specified below). The figures are clear, of very high quality, relevant and adequate, with clear labeling (with one exception, see below Fig 1b) and legends.

We strongly recommend this manuscript, with minor revisions, to be published in Nature Communications.

Thank you very much for a thorough and thoughtful review. We respond to your comments using red text below.

Detailed comments:

L 5: Please name a few of those specific processes that are difficult to capture in the satellite era (examples).

Examples of these processes are covered in Alley et al. (2023). This paper outlines, for example, challenges related to the limited number of observations made to date of collapse-related processes — especially over the timescales that ice sheets equilibrate to such changes. Limited instrumental observations also exist for how ice shelf growth happens. A brief synopsis of these issues is already provided in both the introduction (Lines 18-22; 65-67) and discussion (Lines 243-246) sections of the manuscript. Although the reviewers suggest that we list some of these examples in the abstract, we would prefer to explore this topic in the detail and nuance it deserves within the main body of the manuscript rather than risk diverting away from the main narrative of the abstract, which is already close to the maximum length permitted for this journal.

L 87-88: Please add a short statement about what the overlying 35 m of sediments in the investigated area consists of (glacial marine, and Holocene silt/clay, till?), with reference to Graham et al. (2007) and Graham et al. (2010).

The iceberg ploughmarks documented in this study overprint the Late Weichselian glacial lineations reported by Graham et al. (2007, 2010) that were moulded on top of highly consolidated sediments interpreted as subglacial tills (Sejrup et al., 1994, 2015) and ice-overridden glacimarine sediments (Graham et al., 2010; Sejrup et al., 2015), respectively. These subglacial sediments contain (micro-)fossils radiocarbon dated to ≥ 32 cal. ka BP (Sejrup et al., 1994, 2015; Graham et al., 2010; Reinardy et al., 2025) and are themselves buried by a sequence of proximal and distal glacimarine diamictons, sands and silts with ages predominantly ranging from 17–14 cal. ka BP (Graham et al., 2010; Sejrup et al., 2015; Reinardy et al., 2025). A radiocarbon date from the base of this sequence in a borehole investigated by Graham et al. (2010) constrains the age of the underlying subglacial sediments to >19.5 cal. ka BP. The sequence is overlain by ice-distal shallow marine and marine stratified and laminated muds and silts, whose age has been constrained to ≤ 14 cal. ka BP and thus postglacial and Holocene (Graham et al., 2010; Sejrup et al., 2015).

Following this comment, we have now altered lines 87-88 (now lines 92-95) to include a reference to the lineations being formed at the top of subglacial sediments: “The ploughmarks are distinct from other previously described glacial lineations in the region (e.g., mega-scale glacial lineations, flutes, and megaflores) that were moulded on top of subglacial tills and ice-overridden glacimarine sediments that are older than 19.5 ka (Sejrup et al., 1994; Graham et al., 2007; Graham et al., 2010) (buried ~35 m below seafloor; b.s.f.).

As this section primarily interprets the morphology of the curvilinear grooves, we thought it better to include a statement about the character of the sediments above the ploughmarks in a later section than requested by the reviewers (lines 194-197). This now reads: “At shallower burial depths within our data at this location (~12 m b.s.f.), evidence of multi-keeled iceberg ploughmarks disappears and stratigraphic horizons dated largely to 17–14 ka are instead ploughed by smaller single V-shaped grooves; these in turn are buried by postglacial Holocene muds and silts (Stoker and Long, 1984; Graham et al., 2010).”

L 11: Is there any proxy data or evidence from atmospheric/ climate side that can complement the massive surface melting?

Thank you for this question. For similar reasons as discussed in our response to comment line 5, we suggest that additional discussion related to atmospheric/climatic proxy data is better placed in the main text rather than the abstract. The most relevant section to this question is discussed in lines (233-243), which for balance also includes alternative explanations to the massive surface melting hypothesis, including internal ice sheet dynamics:

“Although reconstructions of the demise of the BIIS show that retreat initiated at around 22 ka — thousands of years prior to deglacial warming and CO₂ rise — changes in ice sheet volume appear to mirror variability in June insolation (Clark et al., 2022b). This suggests that the earlier start of deglaciation may have been triggered by orbitally-induced increasing levels of summer solar radiation that produced greater surface melting (Ryan et al., 2019; Clark et al., 2022b). Accordingly, it is feasible that the magnitude of surface melting reached a threshold between ~19–17 ka, causing the ice shelves fringing the BIIS to undergo widespread hydrofracture and collapse, accelerating ice losses around its marine margins. Alternatively, changes to internal ice sheet dynamics, potentially driven by the effects of sea-level rise and/or atmospheric or ocean warming, could also have altered the calving regime through facilitating different rifting patterns and calving frequencies.”

In response to this comment, we now also refer to recently published sedimentary evidence that further supports our hypothesis: “A shift in surface meltwater presence has also been linked to the rapid deposition of thick grounding-line proximal glacimarine muds that were deposited in the Witch Ground Basin at around this time (Reinardy et al., 2025).”

In addition, in our response to comments on L173-183 we have also cited evidence that most previous disintegration-style events that have been recorded in the satellite era were ultimately triggered by surface melt, which also provides additional support to our interpretation.

L 19 :Comment to the authors /editor about style – using superscript index numbers for references, it looks a bit weird when references are given as examples, “Limited to about 10 e.g., 4,7-11 “. Why not but the “e.g.” in superscript too?

Thank you – this was a typesetting issue related to the referencing software used for the manuscript. We have now corrected this.

L 24-25: Can you please write a sentence how number of pre-existing fractures in calving margin is related to type of ice produced?

This concept forms the argument made by Bassis and Jacobs (2013), and would be difficult to summarise in a single sentence here due to the complexity of the calving process. As discussed by Benn et al. (2007), first-order controls on calving include the strain rate (which arises from spatial variations in velocity, particularly sliding speed); this in turn determines the location and depth of surface crevasses. Second-order processes that can further erode the ice margin include fracture propagation in response to local stress imbalances in the immediate vicinity of the glacier front; undercutting of the glacier terminus by melting at or below the waterline; and bending at the junction between grounded and buoyant parts of an ice tongue. Calving of projecting, submerged ‘ice feet’ can be regarded as a third-order process, because it is paced by first- or second-order calving above the waterline.

Given the complexity of this topic (and the large numbers of processes that would need to be discussed), we feel that a very detailed explanation of how the number of pre-existing fractures in a calving margin is related to the type of iceberg produced would be difficult to incorporate in the introduction without shifting focus away from the main narrative of the paper. Instead, we have therefore focussed on directing readers to other papers which explicitly discuss this topic by modifying the sentence to read: “Observations and numerical models show that the dominant regime of iceberg calving (i.e., the size, shape and number of icebergs calved), and thus the type of iceberg produced, reflects the dynamics of the parent ice mass and the number of pre-existing fractures in a calving margin (cf. Benn et al., 2007; Bassis and Jacobs, 2013).”

L 65: How do you trace their disappearance? I would like to see this later.

This phrasing refers to the switch in the type of iceberg ploughmark observed at different stratigraphic depths in our data. We are happy that the reviewers liked this phrasing, so we have modified lines 194-200 to highlight how we have traced the disappearance of the multi-keeled ploughmarks (and thus the tabular icebergs which produced them):

“At shallower burial depths within our data at this location (~12 m b.s.f.), evidence of multi-keeled iceberg ploughmarks disappears and stratigraphic horizons dated largely to 17–14 ka are instead ploughed by smaller single V-shaped grooves; these in turn are buried by postglacial Holocene muds and silts (Stoker and Long, 1984; Graham et al., 2010). This switch in the type of iceberg ploughmark demonstrates that a calving regime shift from broad parallel comb-like to single iceberg ploughmarks occurred sometime between ~19–17 ka, when the sporadic calving of large tabular icebergs ceased and instead switched to the production of many smaller single-keeled icebergs.”

L 93-95: Is this from some literature or found in the present HR3D survey? In case of former, please provide the reference. In case of latter, add some detail on it.

This statement comes from similar examples found within the HR3D cubes at much deeper depths than can be attributed to the Late Weichselian, and are therefore obviously much older. These examples are less numerous than the examples documented in this paper, which may potentially reflect preservation issues or simply the limited coverage of our HR3D seismic data in this area. For these reasons, we wish to keep the focus of this manuscript on the Late Weichselian examples. In response to this comment, we have reworded

this sentence to make it clearer that we are referring to other examples buried deeper within our own data here:

“Some HR3D seismic datasets also contain examples of similarly ploughed surfaces buried deeper than the Late Weichselian lineations, suggesting that the formative process responsible for the ploughmarks has recurred multiple times during the Quaternary as the North Sea was repeatedly glaciated.”

L 108: The comment about palaeo water depths at 20-18 ka is the first mention of the age (apart from the Abstract, but that should be a standalone text). This is a bit confusing as the discussion about the likely age of the iceberg scours comes later. This statement needs to be preceded by some motivation for the choice of age or be replaced by a more general statement about the likely water depth (e.g. a guess is that the water was deeper than ~20 m (i.e. deeper than sea ice reach) all the time since 30 ka at the locations in question). We also suggest you add a reference to Clark et al., 2022b (<https://doi.pangaea.de/10.1594/PANGAEA.945729>), where the palaeo-bathymetry map datasets are published. It is not clear where you got the numbers ~65-180 m palaeo water depth from. Two references are given (Bradley et al., 2023; Clark et al., 2022) which are correct and relevant in that they describe how the palaeo-bathymetry for the whole BRITICE region was calculated, but the exact information (the palaeo-water depth at your location) is not clearly given in any of these papers.

Thank you for pointing out the inconsistency in the text over when the likely age of the ploughmarks is first introduced. We agree that it is more logical to discuss the likely age of the ploughmarks in a later section where the stratigraphic context of the features can be appropriately discussed (see section: “Did ice shelf disintegration trigger the retreat of the BIIS from the North Sea?”). Therefore, we have chosen to remove specific references to the 20-18 ka age of the ploughmarks in this section and instead now focus more generally on the evidence of water depth.

As you have inferred, we used the palaeo-bathymetry map datasets from Clark et al. (2022b) and Bradley et al. (2023) to assess the former water depths at the ploughmark locations. During the growth and retreat of the BIIS between 31-15 ka, water depths in this area were between ~50-180 m.

To address this comment, we have amended the text to now state the range of water depths at the ploughmark locations over the entire 31-15 ka BIIS growth-retreat cycle: “Palaeo-sea level reconstructions that account for eustasy and isostasy (Clark et al., 2022a; Bradley et al., 2023) demonstrate that water depths in the region of the central North Sea where the multi-keeled grooves are buried were between ~50–180 m during the growth and retreat of the last BIIS (31–15 ka). This suggests that large icebergs, rather than thinner sea-ice floes, were responsible for the seafloor scouring.”

We have also added a reference to Clark et al. 2022b as suggested.

L 121-123: “Conversely, the presence of a bathymetric sill [...] would have prevented large icebergs calved from the FIS [...] from drifting towards the Witch Ground Basin”. Delete “presence of a”, as this gives the impression that “if a sill existed it would have prevented...”. The sill is there, evident in the bathymetry. “Conversely, the bathymetric sill [...] would have...” is clearer.

Done – thank you for this suggestion.

L 166-172: The authors show three different types of ploughmarks in Fig. 2 and associate them with different dimensions of icebergs. While an analogy is provided that big tabular iceberg formation was followed by smaller icebergs, this can be further complemented with the location of the different ploughmarks in the direction of ice sheet retreat. Can you correlate the position of ploughmarks and dimension of iceberg with the retreat of the BIIS? We feel this has been attempted in the first paragraph of the third discussion point, but the evidence for this development is not clear. Please clarify this in the text and by adding appropriate notations/labels to Fig. 6 to show where smaller iceberg scours were identified.

Thank you for this suggestion. The orientation of multi-keeled iceberg ploughmarks, where present, and their relation to the ice margin positions of the BIIS during its retreat are displayed in Figs. 1a and 6. The multi-keeled ploughmarks generally exhibit a NW-SE orientation in the more western site locations and transition

towards a more NE-SW orientation as they drift towards the northeast; these observations informed the drawing of the semi-transparent white arrows in Fig. 6 which speculate as to their past drift direction. In contrast, the orientations of the smaller ploughmarks are highly chaotic and generally lack a discernible dominant orientation, especially given the relatively small areal coverage of the HR3D seismic data. This is demonstrated in the rose plots in Fig. 3. A much larger areal coverage of HR3D seismic data would be needed to accurately assess single ploughmark orientation, especially given the contrast between the number of ploughmarks visible in HR3D seismic data compared to conventional 3D seismic data (Fig. 3).

Although we are able to stratigraphically constrain the timing of the multi-keeled iceberg ploughmarks to sometime between the retreat of grounded ice from the central North Sea around ~19–17 ka (based on the fact that multi-keeled ploughmarks disappear at shallower depths and are replaced by smaller single-keeled ploughmarks), advances in the resolution and precision of chronological constraints in the North Sea would be required to date the age of the ploughmarks more precisely than is possible presently. Therefore, it is difficult to correlate the position of ploughmarks to a specific ice margin with more precision than has already been attempted.

In response to this comment, we have modified the manuscript in two ways:

- 1) We have noted in the text that the multi-keeled ploughmarks are only observed within specific surveys that are located in the deeper waters of the Witch Ground Basin and west of the Norwegian Channel (lines 113-114), whereas single-keeled ploughmarks are present in all of the HR3D seismic datasets examined (LINES 108-109). Combined with the response to L65 above, these clarifications should make it clearer that evidence for multi-keeled iceberg ploughmarks disappears and is replaced by single-keeled ploughmarks as the BIIS retreated across the central North Sea.
- 2) Following this comment, we also investigated whether it was possible to link the width of the multi-keeled ploughmarks observed with a particular ice margin based on their dimensions and stratigraphic position, and whether we could proportionally scale the orientation arrows in Figs. 1 and 6 according to the dimensions of the ploughmarks. However, whilst attempting to do this, we came across several issues which mean we have concluded that this change would be difficult to implement and, in the end, would not improve the manuscript. This is because no obvious correlation between geographic location and iceberg width could be discerned; this likely results from the ploughmarks being formed either at different times (different calving margins) or the individual shapes of the iceberg keels changing at different points of their lifecycle (i.e. the width of the ploughmark may not be necessarily directly proportional to iceberg width as the footprint of the keel may change depending on the geometry of the icebergs during drift and disintegration). Secondly, several sites contain multiple multi-keeled iceberg ploughmarks of varying width at slightly different depths, and this would be difficult to plot in a meaningful way on Figs. 1a or 6. Instead, we have amended the caption for Fig. 6 to emphasise that the proposed drift directions are inferred from the orientation of the multi-keeled iceberg ploughmarks: “Proposed drift direction of icebergs calved from the BIIS during the last glaciation based on the orientation of multi-keeled iceberg ploughmarks.” We have also added a sentence in LINES (133-137) describing the shift in ploughmark orientation as the icebergs drifted further east: “This pattern of currents would have likely taken icebergs calved from the BIIS in the Witch Ground Basin eastwards towards the Norwegian Channel before they were transported northwards; this proposed drift direction is supported by an observed shift in general ploughmark orientation from NW-SE in the Witch Ground Basin towards a more NE-SW orientation in the HR3D datasets situated further to the east (Figure 6).”

Fig. 1b: The blue colour of the North Sea under the white ice gives the impression that ice was not grounded there at 22-21 ka (which it clearly was). This might be confusing to some readers. I suggest you indicate the present sea level with a thin line across the section and make the North Sea be white (as the ice) with blue hatching (////) below sea level. Also, indicate that the y-axis scale is based on present-day topography.

Thank you for this suggestion. We have removed the filled blue colour of the North Sea and have replaced it with a light blue line labelled 'Modern sea level'. We have now also noted in the revised figure caption that the palaeo topographies are in reference to modern sea level. We have also added labels for the Witch Ground Basin (WGB) and the Norwegian Channel (NC) to the inset in Fig. 1a and describe these abbreviations in the caption.

L173-183: Iceshelf calving can happen irrespective of warming due to changes in stress regime, while warmer climate increase the frequency. For example, due to topographic undulation or some other reason, there can be rift or fracture formation that may lead to calving. This is in particular true for Antarctica and Greenland, but may not be true for BIIS and FIS confluence zone. Basically, ice shelves can break during growing, retreat and intermittent phases. A question in that regard: can you discern that those ploughmarks were due to the icebergs that calved due to warming but not due to "natural" calving or during the growth of ice sheet? Such a clarification would strengthen the text.

This is an interesting comment that is difficult to fully address here with the presently available methods. To discern this further, future research could, for example, apply a more sophisticated calving model for the BIIS with a higher temporal resolution (<10-100 year time steps) than available presently and examine the frequency distribution of tabular icebergs formed in growth vs retreat cycles, or gather additional sediment cores from the locations where multi-keeled iceberg ploughmarks are present in an attempt to more accurately ascertain their age.

The most robust and simple argument that we can present in this manuscript is based on stratigraphic context. As discussed in the manuscript, the multi-keeled ploughmarks are situated above glacial lineations formed by the flow of grounded ice just prior to the retreat of the last BIIS from this part of the central North Sea at ~20 ka (Sejrup et al., 1994; Graham et al., 2010). Therefore, prior to the formation of the ploughmarks, the ice sheet was grounded and tabular icebergs were calved sporadically from relatively stable ice shelves (in a configuration similar to that described in the introduction). As these multi-keeled ploughmarks are overlaid by more numerous single-keeled ploughmarks, the simplest and most likely interpretation is that this transition reflects the collapse of these ice shelves during the retreat phase of the ice sheet.

To address this comment, we have added an additional paragraph at the start of the third discussion section. This text directly acknowledges that the rate of iceberg calving is primarily set by the stress regime, but then continues to lay out how stable ice shelves can undergo calving regime transitions due to substantial changes in stress regime or environmental factors which can drive rapid changes to terminus position. This section now reads:

"Modern rates of iceberg calving are primarily set by structural and internal stresses within an ice shelf created as ice flows towards the ice margin (Benn et al., 2007; Alley et al., 2023; Walker et al., 2024). Increased ice shelf fracturing or thinning due to substantial changes in ice flow or ocean-induced basal melting can diminish the structural integrity of an ice shelf, potentially resulting in a calving regime transition that can drive rapid changes to terminus position, or make an ice shelf more susceptible to external environmental drivers that could precondition or trigger rapid collapse (cf. Alley et al., 2023; Walker et al., 2024). Most rapid disintegration-style events observed to date in the observational era were ultimately triggered by enhanced surface and/or basal melting (Vaughan and Doake, 1996; Scambos et al., 2000; MacAyeal et al., 2003; Scambos et al., 2004; Arthur et al., 2021; Millan et al., 2023)."

We believe that this clarification strengthens the stratigraphic argument outlined in subsequent paragraphs, which postulates that the most likely interpretation is that the transition in ploughmark type observed in our data reflects ice shelf collapse during the retreat phase of the BIIS.

L 388, 391, 526, 533, 537: doi:doi: Remove "doi:".

This was a referencing software issue and has now been corrected.

Fig. 5. Please indicate North (with an arrow or in the caption) in the HR3D images. Please add the locations of these in Fig. 6. Why not show the fifth example?

We now indicate the direction of north in this figure as requested (see each greyscale interpretation panel).

We have chosen to prioritise showing the most easily visible examples of multi-keeled ploughmarks in this figure rather than choosing one from each cube because examples from some cubes (e.g. 5a, which can even be mapped as a seismic horizon rather than just a timeslice) are far easier to distinguish than in others due to variations in HR3D survey quality owing to differing acquisition conditions. Examples B and D are actually from the same cube due to the high quality of this particular dataset, for example. To make this distinction clearer to the reader, we have now rephrased the caption to read: “Selected multi-keeled iceberg ploughmarks buried beneath the North Sea, imaged in HR3D seismic data.”

We have also added an additional example from the easternmost HR3D seismic cube (new panel (e)) in response to the comment below on Fig. 6.

Fig. 6. Please indicate where the HR3D-slices from Fig. 4 and 5 are located in the map (maybe by a small index number next to the arrow-circles). It would be especially interesting to see which ploughmark set that is the easternmost indication in Fig. 6 (possibly calved from the FIS).

We have labelled the locations of the HR3D slices shown in Fig. 5 as small numbers. As discussed in our response to the comment on Fig. 5, not all examples from all HR3D cubes are shown in Fig. 5 because of variation in quality/ease of distinction. However, we have added the example from the easternmost site as requested as new panel (e).

The examples shown in Fig. 4 are from the Weddell Sea and Barents Sea, as indicated in the caption for this figure.

Methods: Please include information about what frequency span the hydrophones were sensitive to.

We have changed “20–250 Hz dominant signal frequency” to “20–250 Hz effective frequency band” as in the previous version slightly incorrect terminology was used for the dominant frequency. Dominant frequency usually refers to the peak or centre frequency used to estimate seismic resolution, which in our case appears to be about 115 Hz based on a 4 m vertical resolution and 1850m/s sound velocity.

We have also added the following information into the methods in regard to the comment on hydrophone frequency span: “The data were acquired using two 1200 m long streamers towed 3 m beneath the sea surface with 96 hydrophone groups at 12.5 m spacing with a response that varied by $\leq \pm 0.5$ dB over frequencies between 2–500 Hz, a 6.25 m shot interval and a 1-ms sample rate”.

Richard Gyllencreutz and Ankit Pramanik

Reviewer 3:

[Our response to Reviewer 3 also includes those comments annotated in the text sent back to the authors along with the revision; our responses to these annotated comments are incorporated in the replies below]

What are the noteworthy results?

The noteworthy results of this study are applying High-resolution 3D (HR3D) seismic data interpretations to the North sea to document and interpret curvilinear grooves with three distinct morphologies to better understand the evolution of the British-Irish Ice Sheet. This new approach using HR3D seismic data allowed the research team to identify three distinct features that are compared to the Barents, and Weddell Sea multi-keeled ploughmarks. Additionally, this data is compared to previous studies in the region to develop a timeline of events, and to better assess the role of ice shelves in promoting stability in the British-Irish Ice sheet during early deglacial warming. Furthermore, these observations have the potential to better assess polar response to warming, particularly in or around Antarctica as there are many similarities between the two environments described by the team.

Will the work be of significance to the field and related fields? How does it compare to the established literature? If the work is not original, please provide relevant references.

This work is significant in that it provides an opportunity to study the transition in calving regime of Northern Hemisphere ice sheets and the loss of ice shelves documented in this study can precondition or even start the demise of an ice sheet, which is critical for understanding how the Antarctic Ice Sheet will respond to warming temperatures. These observations are some of the missing critical pieces needed to build confidence in our projections of future sea level rise from the loss of ice sheets.

Does the work support the conclusions and claims, or is additional evidence needed?

The authors do a good job of supporting their conclusions and claims. Providing evidence of similar features and a comprehensive description of the prior data in this region is organized in a way that steps through the new technique, shows how it improves on previous seismic observations, and then applies the new data from HR3D seismic to the environment to get a sense of the ice sheet/shelf history in the North sea and compares to other observations in the Barents and Weddell sea.

Are there any flaws in the data analysis, interpretation and conclusions? Do these prohibit publication or require revision?

The authors do an excellent job of presenting their data and interpreting it based on other observations in the North Sea and other environments past and present that have similar features, trying to better understand this complex system.

Is the methodology sound? Does the work meet the expected standards in your field?

The methodology and conclusions seem to build off of the literature and provide evidence for how this new technique is important for a more detailed look at these important settings. My field is not in seismic interpretation, so it is hard for me to fully say, but I do work in the subglacial environment, and I believe their conclusions fit the data and interpretations they presented.

Is there enough detail provided in the methods for the work to be reproduced?

My background is not in seismic interpretations, but it seems that they provide robust references and descriptions for how they collected, processed, and interpreted the data.

Thank you very much for a thorough and thoughtful review. We reply to your comments below – our answers also include replies to comments annotated in the copy of the manuscript we received back after review.

Other minor comments highlighted in the text:

Line 18: ice-shelf from my understanding should not be hyphenated. There are times in the text where it is and so I would say for consistency just make sure you are either hyphenating all the ice shelf/shelves or removing it hyphen.

We have removed instances where ice shelf is hyphenated throughout the text.

Line 21: ice sheet (similar to comment for line 18)

We have removed instances of hyphenation.

Line 22: In your call out to ice shelves being vulnerable to changes in atmospheric and ocean conditions you cite a paper on annual mass budget, which uses this statement as well. Might be better here to cite papers documenting atmospheric and ocean conditions leading to ice shelf vulnerability.

Thank you for this comment. We have replaced the previous reference with a selection of others (Pritchard et al., 2012; Banwell et al., 2013; Massom et al., 2018; Walker et al., 2024) that between them document the range of atmospheric and oceanic drivers that can lead to ice shelf vulnerability.

Line 28: Seems like there should be some citations here of sporadic tabular icebergs followed by periods of quiescence between calving.

We have added three citations to this sentence that describe the observed calving frequency of large tabular icebergs from Antarctic ice shelves (Lazzara et al., 1999; Fricker et al., 2002; Hogg and Gudmundsson, 2017).

Line 51-52: This sentence seems a bit awkward, so it might be good to rewrite/restructure it.

Thank you for this feedback. We have reworded this sentence to now read: "... its predictions are supported by the presence of grounding-zone wedges (GZWs; submarine glacial landforms associated with ice shelf presence), particularly around the BIIS' western margin".

Line 61-64: The sentence seems a bit awkward. It might be better to say, ".. North Sea using newly captured high-resolution three-dimensional (HR3D) seismic data to infer.."

We have altered the wording of this sentence to clarify here; however, it is the morphology of the iceberg scours that is newly captured (the HR3D seismic data are also fairly recent, but this is not what was intended to be emphasised here). We have therefore deleted "newly captured by" in order to avoid this confusion.

Line 73: This might be too picky (and you might not have the words/characters) but it would be good to introduce these features. "We identify three classes of features associated with curvilinear grooves from HR3D seismic in the North Sea: the first feature...."

We would prefer not to make this alternation as we feel that the first sentence of this section already summarises this point and explains that the features have three distinctive morphologies:

"Inspection of HR3D seismic data in seven different survey areas of the central North Sea reveals an abundance of curvilinear grooves with three distinctive morphologies".

We therefore wish to avoid duplicating this point shortly after the first sentence of this section.

Line 97-99: It would be good to cite some of the work that supports the individual iceberg for these characteristic scours. If they are distinguishing features and it has been observed in other regions/associated with individual icebergs then you should cite those studies like the other classes have specific citations.

The morphology of the individual ploughmarks identified in the HR3D seismic data are near-identical to those interpreted as iceberg ploughmarks from high-latitude continental shelves and submarine ridges in many areas of the Arctic and Antarctic seas and we have therefore cited three additional references here (listing them as example references as a full list would be too extensive here). These are: (Woodworth-Lynas et al., 1991; Dowdeswell et al., 1993; Dowdeswell and Ottesen, 2024).

Line 135: Add hyphen: palaeo-record

Added.

Line 157: Add hyphen: palaeo-water

Added.

Line 208: should there be a hyphen?

Removed.

Line 218: should there be a hyphen?

Removed.

References

1. Alley, R. B., Cuffey, K. M., Bassis, J. N., Alley, K. E., Wang, S., Parizek, B. R., Anandakrishnan, S., Christianson, K., and DeConto, R. M. (2023). Iceberg Calving: Regimes and Transitions. *Annual Review of Earth and Planetary Sciences*, v. 51, no. 1, 189-215, <https://doi.org/10.1146/annurev-earth-032320-110916>.
2. Arthur, J. F., Stokes, C. R., Jamieson, S. S. R., Miles, B. W. J., Carr, J. R., and Leeson, A. A. (2021). The triggers of the disaggregation of Voyeykov Ice Shelf (2007), Wilkes Land, East Antarctica, and its subsequent evolution. *Journal of Glaciology*, v. 67, no. 265, 933-951, <https://doi.org/10.1017/jog.2021.45>.
3. Banwell, A. F., MacAyeal, D. R., and Sergienko, O. V. (2013). Breakup of the Larsen B Ice Shelf triggered by chain reaction drainage of supraglacial lakes. *Geophysical Research Letters*, v. 40, no. 22, 5872-5876, <https://doi.org/https://doi.org/10.1002/2013GL057694>.
4. Bassis, J. N., and Jacobs, S. (2013). Diverse calving patterns linked to glacier geometry. *Nature Geoscience*, v. 6, 833-836, <https://doi.org/10.1038/ngeo1887>.
5. Benn, D. I., Warren, C. R., and Mottram, R. H. (2007). Calving processes and the dynamics of calving glaciers. *Earth-Science Reviews*, v. 82, no. 3, 143-179, <https://doi.org/https://doi.org/10.1016/j.earscirev.2007.02.002>.
6. Dowdeswell, J. A., and Ottesen, D. (2024). Iceberg ploughmarks and glacial lineations on Jan Mayen Ridge: Evidence for past iceberg and possible ice-shelf grounding in deep water. *Quaternary Science Reviews*, v. 338, 108838, <https://doi.org/https://doi.org/10.1016/j.quascirev.2024.108838>.
7. Dowdeswell, J. A., Villinger, H., Whittington, R. J., and Marienfeld, P. (1993). Iceberg scouring in Scoresby Sund and on the East Greenland continental shelf. *Marine Geology*, v. 111, no. 1, 37-53, [https://doi.org/https://doi.org/10.1016/0025-3227\(93\)90187-Z](https://doi.org/https://doi.org/10.1016/0025-3227(93)90187-Z).
8. Fricker, H. A., Young, N. W., Allison, I., and Coleman, R. (2002). Iceberg calving from the Amery Ice Shelf, East Antarctica. *Annals of Glaciology*, v. 34, 241-246, <https://doi.org/10.3189/172756402781817581>.
9. Hogg, A. E., and Gudmundsson, G. H. (2017). Impacts of the Larsen-C Ice Shelf calving event. *Nature Climate Change*, v. 7, no. 8, 540-542, <https://doi.org/10.1038/nclimate3359>.
10. Lazzara, M. A., Jezek, K. C., Scambos, T. A., MacAyeal, D. R., and van der Veen, C. J. (1999). On the recent calving of icebergs from the Ross Ice Shelf 1. *Polar Geography*, v. 23, no. 3, 201-212, <https://doi.org/10.1080/10889379909377676>.
11. MacAyeal, D. R., Scambos, T. A., Hulbe, C. L., and Fahnestock, M. A. (2003). Catastrophic ice-shelf break-up by an ice-shelf-fragment-capsize mechanism. *Journal of Glaciology*, v. 49, no. 164, 22-36, <https://doi.org/10.3189/172756503781830863>.
12. Massom, R. A., Scambos, T. A., Bennetts, L. G., Reid, P., Squire, V. A., and Stammerjohn, S. E. (2018). Antarctic ice shelf disintegration triggered by sea ice loss and ocean swell. *Nature*, v. 558, no. 7710, 383-389, <https://doi.org/10.1038/s41586-018-0212-1>.
13. Millan, R., Jager, E., Mouginot, J., Wood, M. H., Larsen, S. H., Mathiot, P., Jourdain, N. C., and Bjørk, A. (2023). Rapid disintegration and weakening of ice shelves in North Greenland. *Nature Communications*, v. 14, no. 1, 6914, <https://doi.org/10.1038/s41467-023-42198-2>.
14. Pritchard, H. D., Ligtenberg, S. R. M., Fricker, H. A., Vaughan, D. G., van den Broeke, M. R., and Padman, L. (2012). Antarctic ice-sheet loss driven by basal melting of ice shelves. *Nature*, v. 484, no. 7395, 502-505, <https://doi.org/10.1038/nature10968>.
15. Scambos, T. A., Bohlander, J. A., Shuman, C. A., and Skvarca, P. (2004). Glacier acceleration and thinning after ice shelf collapse in the Larsen B embayment, Antarctica. *Geophysical Research Letters*, v. 31, no. 18, <https://doi.org/https://doi.org/10.1029/2004GL020670>.
16. Scambos, T. A., Hulbe, C., Fahnestock, M. A., and Bohlander, J. (2000). The link between climate warming and break-up of ice shelves in the Antarctic Peninsula. *Journal of Glaciology*, v. 46, 516-530, <https://doi.org/10.3189/172756500781833043>.
17. Vaughan, D. G., and Doake, C. S. M. (1996). Recent atmospheric warming and retreat of ice shelves on the Antarctic Peninsula. *Nature*, v. 379, no. 6563, 328-331, <https://doi.org/10.1038/379328a0>.
18. Walker, C. C., Millstein, J. D., Miles, B. W. J., Cook, S., Fraser, A. D., Colliander, A., Misra, S., Trusel, L. D., Adusumilli, S., Roberts, C., and Fricker, H. A. (2024). Multi-decadal collapse of East Antarctica's Conger–Glenzer Ice Shelf. *Nature Geoscience*, v. 17, no. 12, 1240-1248, <https://doi.org/10.1038/s41561-024-01582-3>.

19. Woodworth-Lynas, C. M. T., Josenhans, H. W., Barrie, J. V., Lewis, C. F. M., and Parrott, D. R. (1991). The physical processes of seabed disturbance during iceberg grounding and scouring. *Continental Shelf Research*, v. 11, no. 8, 939-961, [https://doi.org/https://doi.org/10.1016/0278-4343\(91\)90086-L](https://doi.org/https://doi.org/10.1016/0278-4343(91)90086-L).